# Assemblages of Meiobenthic and Planktonic Microcrustaceans (Cladocera and Copepoda) from Small Water Bodies of Mountain Subarctic (Putorana Plateau, Middle Siberia)



Elena S. Chertoprud [1,2,*], Anna A. Novichkova [1,2], Aleksandr A. Novikov [3], Elena B. Fefilova [4], Lada V. Vorobjeva [5], Dmitry S. Pechenkin [5] and Aleksandr I. Glubokov [5]

1   Biological Faculty, Moscow State University, Leninskie Gory, 119991 Moscow, Russia; anna.hydro@gmail.com
2   Severtsov Institute of Ecology & Evolution, Leninsky Pr. 33, 119071 Moscow, Russia
3   Department of Zoology and General Biology, Institute of Fundamental Medicine and Biology, Kazan Federal University, 18 Kremlyovskaya, 420008 Kazan, Russia; aleksandr-novikov-2011@list.ru
4   Institute of Biology, Komi Scientific Centre, Ural Branch of Russian Academy of Sciences, 28 Kommunisticheskaya, 167982 Syktyvkar, Russia; fefilova@ib.komisc.ru
5   Russian Federal Research Institute of Fisheries and Oceanography, 17 V. Krasnoselskaya, 107140 Moscow, Russia; vorobjeva.lada@yandex.ru (L.V.V.); pechenkinds@gmail.com (D.S.P.); glubokov@vniro.ru (A.I.G.)
*   Correspondence: horsax@yandex.ru; Tel.: +7-9163444396

**Abstract:** The Putorana Plateau (Krasnoyarsk Territory, Russia) is one of the largest mountainous regions of subarctic Eurasia. Studies of aquatic ecosystems of this are far from complete. In particular, microcrustaceans (Cladocera и Copepoda) of the Putorana Plateau are poorly investigated, although they are one of the main components of meiobenthic and zooplanktonic communities and a target for monitoring of the anthropogenic influence and climate change. An open question is a biogeographical status of the crustacean fauna of the plateau. Additionally, it is unknown which environmental factors significantly affect benthic and planktonic crustacean assemblages? Based on the samples collected in tundra and forest tundra ponds in the western and central parts of the plateau, analysis of the composition of crustacean fauna and factors regulating the assemblage structure was performed. In total, 36 Cladocera and 24 Copepoda species were found. Of these, 23 taxa are new for the region, and four are new to science. Species richness of Copepoda is higher in the central part and on the western slopes of the plateau than in foothills, while number of the Cladocera species in contrast decreases in mountainous areas. Variations in meiobenthic assemblages are due to the research area, type of water supply and less affected by altitude above sea level. For planktonic assemblages the size of the water body and, to a lesser degree, by macrophytes species composition was significant. Almost 12.8% of microcrustacean species of the Putorana Plateau can be attributed to glacial relics. Crustacean fauna of the Putorana Plateau has a high species richness and distinguishes significantly from the fauna of both western and eastern regions of the Arctic. The specifics of faunal composition of the region are connected to the climatic features of Middle Siberia and the retaining of the Pleistocene fauna in some glacial refugia.

**Keywords:** Putorana Plateau; zooplankton; meiobenthos; Cladocera; Copepoda; ecological factors; biogeography

## 1. Introduction

The Putorana Plateau (Krasnoyarsk Territory, Russia) is one of the largest mountainous regions of the subarctic zone of Eurasia [1]. Its landscape is characterized by elongated large lakes lying in deep valleys. The plateau mountains do not have rocky peaks and at an altitude of about 1000 m a.s.l. turn into flat flat-topped interfluve [2]. Although the Putorana Plateau lies beyond the Arctic Circle and its climate is harsh [3], glaciological and geomorphological studies indicate that the last glaciation did not cover this area

entirely [4,5]. During the period when the thickness of the glacial icecaps at the top of the plateau reached several hundred meters, only several glacier tongues went down the valleys [4]. This indicates the possibility of retaining of glacial relics in the reservoirs of the plateau and the potentially high species richness of the fauna [6,7].

Data on the composition of aquatic invertebrates of the Putorana Plateau are scarce, due to its inaccessibility [8]. The first brief data on the plankton microcrustaceans (Cladocera and Copepoda) of Lake Lama was obtained in 1937 during an expedition of the Institute of Polar Agriculture, Animal Husbandry and Fisheries [9]. Later studies on the zooplankton composition were carried out for lakes Lama, Glubokoe, Sobachye, Keta, Kutaramakan [10,11], the Khantai reservoir [12,13] and some reservoirs of the river basins Nyakshinda, Dupkun and others [14,15]. The listed publications are devoted primarily to the fauna of large lakes important for fishing. Studies of the fauna of small reservoirs are mainly confined to the western slopes of the plateau [16]. For a series of lakes, the composition of microcrustaceans in water bodies with different water acidity is described. It has been shown that within the plateau a longitudinal height-related gradient of variability of zooplankton species composition is more pronounced than the latitudinal one [16]. Comparison of the Copepoda fauna of the Putorana Plateau lakes with the Bolshezemelskaya tundra (Western Siberia) revealed that the species richness of Calanoida is higher in the eastern regions than in the western ones [17]. Data on the potential southern invasive species has been obtained, and it is assumed that in the event of a warming climate in the region, the species richness and abundance of Cladocera will increase [16]. Based on the summarized results of all publications, list of 102 taxa of Cladocera and Copepoda of the Putorana Plateau were compiled [18,19].

However, many questions regarding the composition and the structure of the microcrustacean assemblages of the Putorana Plateau waters remain open up to day. Is the fauna of the region rich? Does it include endemic and relict species, or impoverished as it is typical for polar regions? What biogeographic status does the territory of the Putorana Plateau have? How different are the assemblages of Cladocera and Copepoda in reservoirs of different formation types? Finally, what environmental factors are drivers for planktonic and benthic crustaceans in the mountainous Subarctic?

The current investigation, based on extensive material from the central and western part of the Putorana Plateau, focused on comparative analysis of the structure and factors regulation of planktonic and meiobenthic microcrustaceans (Cladocera and Copepoda) assemblages in the small lakes. In parallel, a detailed taxonomic analysis of the revealed species was performed, which made it possible to assess the biogeographic traits of the region. The novelty of this study is that comprehensive studies of aquatic communities of the mountainous polar regions of Eurasia, covering simultaneously different taxonomic and ecological (plankton, benthos) groups of crustaceans, are rare. At the same time, alpine water ecosystems of high latitudes are recognized as the most important objects for monitoring and observation of anthropogenic impact and global climate change [20].

## 2. Materials and Methods

### 2.1. Studied Area

Studies were performed in central and western parts of the Putorana Plateau, including flat foothill areas in August of 2021 during the expedition of Moscow State University together with the stuff of the nature reserves of Taimyr Reserves. Small water bodies of the basins of lakes Ayan, Kutaramakan and Keta, as well as lakes of the river valleys Neral, Burgul, Irkinda and Rybnaya were sampled (Figure 1). The height difference between the highest mountainous and lowland sampling stations was 463 m (from 53 to 516 m a.s.l.), the maximum distance between the futhest stations was about 200 km.

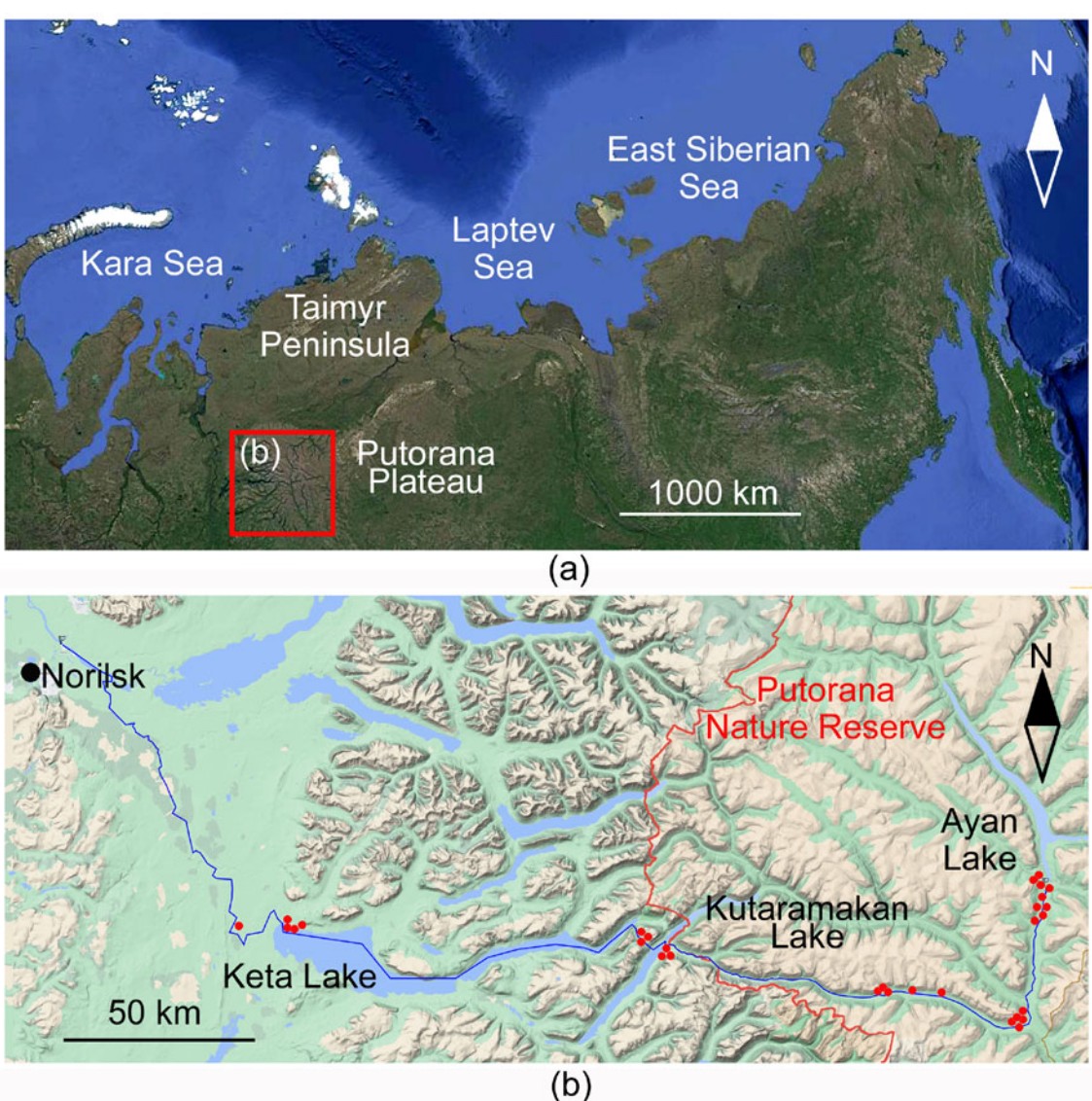

**Figure 1.** Map of Middle and Eastern Siberia (**a**) with position of the Putorana Plateau (red square); Putorana Plateau (**b**) with location of sampling stations (red points), border of Putorana Nature Reserve (red line) and expedition track (blue line).

The explored region of the Putorana Plateau is characterized by an Arctic continental climate with an average annual temperature of approximately −10.2 °C and average annual precipitation of 450–700 mm [3]. The winter period lasts for about six months from October to March and characterized by average temperatures of −28–30 °C [21]. Summer lasts for three months, when the average temperature of 11–13 °C, but the maximum can reach up to 30 °C. The depth of permafrost on the Putorana Plateau in the summer varies from 0.5 to 1.5 m [3]. The vegetation of the central part of the plateau in the area of Lake Ayan and in the upper reaches of Neral River was represented with typical shrub tundra, turning into larch forest tundra in the lowlands and on the western slopes [22].

*2.2. Types of Waterbodies*

In total, zooplankton and meiobenthos of 30 water bodies typical of a mountainous tundra and lowland forest tundra were studied. The following types of water bodies have been studied: low flow lakes, surving as outflows or formed during the expansion of the stream and riverbed (Figure 2a,b); oxbows with constant or drying during the summer connection to the river course (Figure 2c); swampy lakes located on raised moss

bogs (Figure 2d); drainless, including mountainous, lakes lying in the relief depressions (Figure 2e,f). The size of the reservoir ranged from 600 m$^2$ to 7 km$^2$.

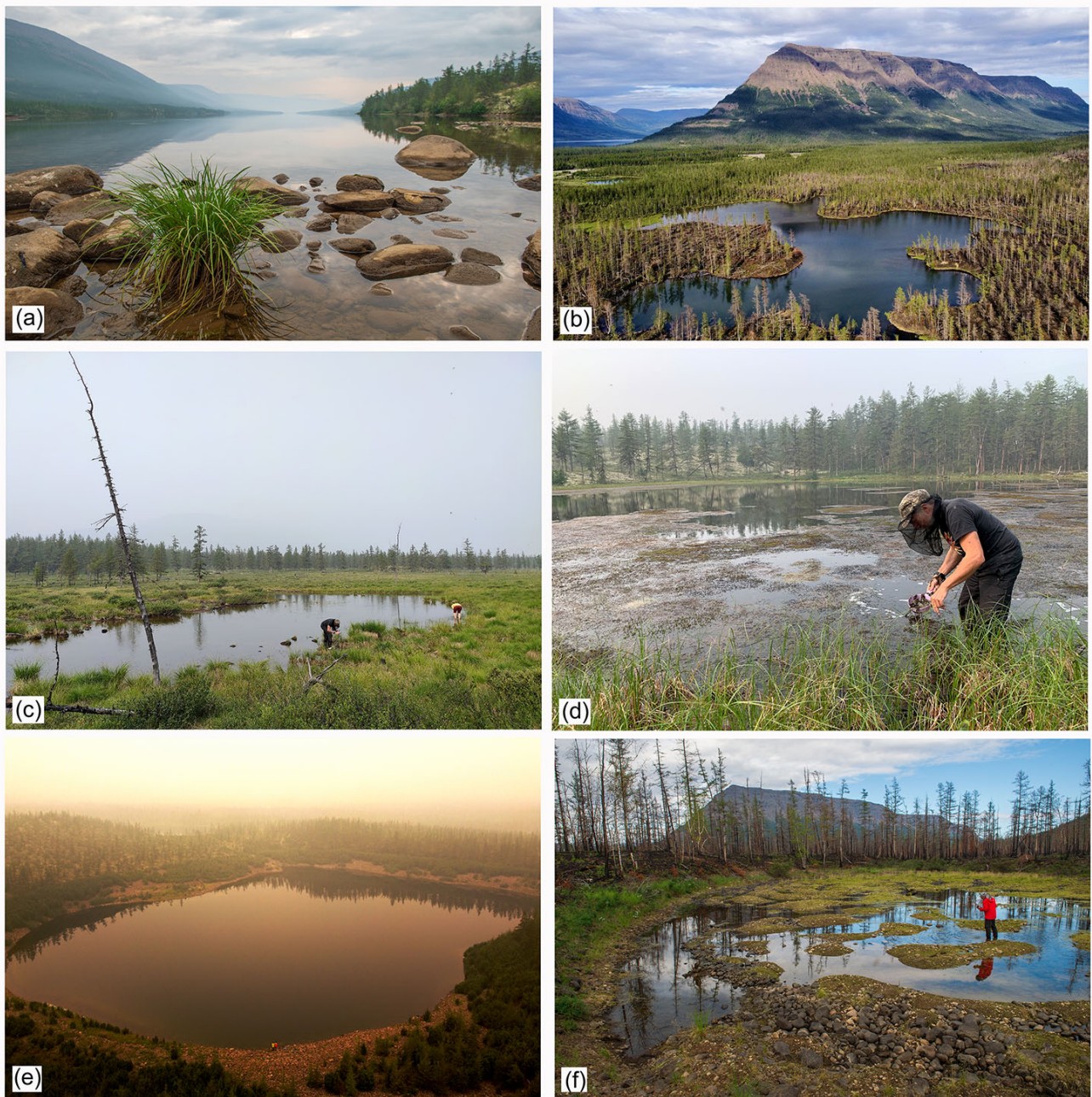

**Figure 2.** Main types of water bodies on Putorana Plateau: Lake Manumakly source of the Ikinda River (**a**); flowing lake on plain (**b**); oxbow (**c**); swampy lake with moss floating fen (**d**); endorheic lake in a rocky basin (**e**); drying up endorheic lake (**f**). (Photos authors: S.V. Vakorin—(**a**,**b**,**d**,**f**); I.P Sadchikov—(**c**,**e**)).

There is a trend of shift of water bodies characteristics from mountain areas with a predominance of drainless lakes in rocky basins to flat floodplains in the lower reaches of rivers with swamp lakes and oxbows was noted. A similar change in hydrological types of reservoirs is typical both for the Putorana Plateau and the mountainous regions of other latitudinal zones [3].

## 2.3. Sampling

At each site, quantitative samples of zooplankton were collected by hauling a plankton net (diameter 0.1 m, 50 μm mesh) horizontally through the water column parallel to the bottom. The volume of the filtered water was calculated based on the length of the net path through the water, measured at each site. Three replicates were taken at each station and combined into one mixed sample afterwards. The volume of each mixed sample was 48–50 L. The meiobenthos was sampled using a plastic tube that was inserted into the uppermost 3–4 cm of the sediment layer. From each site, three substrate portions were taken randomly, all representing different meiobenthic habitat substrates if possible, and then pooled. Each mixed sample covered an area of 9.4 cm$^2$. The samples were preserved with 96% ethanol and filtered (50-μm mesh) before identification. All the samplings were performed from the shore. In total, 30 mixed samples of zooplankton and 30 samples of meiobenthos were taken.

Altitude, size, and shape were estimated with Garmin Etrex 30 GPS navigator for each water body. A flowage rate was determined with a hydrometric turntable (ISP-1M). At each station, environmental variables such as water temperature, pH, and total mineralization (ppm) were measured with a Yieryi portable multifunctional electronic water quality tester (five in one). Dominant of type of the lake water supply (rain, spring, river) was determined based on a detailed visual inspection of the water body, flow rate characteristics and hydrochemical characteristics of the waters. The type of bottom sediments and the composition of macrophytes were described at each station.

Preliminary species identification and counts were carried out in Bogorov counting chambers. The total numbers of Cladocera (i.e., Anomopoda), Copepoda and Anostraca were recorded. Copepodite stages of Cyclopoida and Calanoida were counted separately but only to the genus level without species identification. An Olympus CX-41 high-power microscope was used for accurate crustacean identification following both standard taxonomic treatises and recent taxonomic revisions: [23–28] for Copepoda; [29–32] for Cladocera; and 23 for Anostraca.

## 2.4. Literature Data

The compiled species lists from the existing database on microcrustaceans [33] were used in the analysis of biogeographical status of fauna of the Putorana Plateau. Additionally, published data on Cladocera and Copepoda species richness throughout Polar zone of the Kola Peninsula, Far East, West, Middle and East Siberia [17,18,34–41] and other were used in the compilation of microcrustacean distribution ranges database. Some parts of relatively old articles cited in a general review by Fefilova et al. [18] are not separately presented in the reference list. Comparison of species lists from different regions was the basis of the biogeographical analysis.

## 2.5. Statistical Analysis

To evaluate the effects of environmental factors on the crustacean community, we used distance-based linear modelling (DistLM) in PRIMER [42]. The test was used to estimate the influence of environmental factors on species richness and general abundance in the observed water bodies. The environmental data involved 12 variables: DIST—district of the research; TYPE—type of the water body; FEED—type of water supply; AREA—total area of the water body; ALTIT—altitude above the sea level; FLOW—flow rate; TEMP—temperature of water, °C; PPM—total mineralization; PH—pH; MACR—dominant macrophyte species in the water body; SEDIM—type of bottom sediments; FISH—absence/presence of fish in the water body. First, marginal tests were performed to determine the effect of each variable on the variation in species assemblage structure. Then, the best-fitting model was selected using the Akaike information criterion AICc. This criterion was used to select significant factors in the model, taking into account sample size by increasing the relative penalty for model complexity with small data sets. Sequential

tests are provided for each variable that is added to the model. A dbRDA (distance-based redundancy analysis) analysis was used to ordinate the fitted values from a given model.

The SIMPER procedure was used to identify the species that made the greatest contribution to the pattern of similarity and dissimilarity in assemblages of meiobenthic and planktonic crustaceans from water bodies with different environmental conditions. The significance of the of sample groups identification with respect to each of the factor was determined based on the ANOSIM procedure. For each sample group, characteristic (contributing most to the similarity between samples of the same group) and differentiating (contributing most to the dissimilarity between samples of different groups) taxa are identified.

We also applied a cluster analysis, to illustrate the comparative analysis of the microcrustacean species composition in northern regions of Eurasia, based on our original data and available literature sources, in PAST [43] (stratigraphic constraints, paired group algorithm, Kulczynsky similarity index). The following regions were considered: Kola Peninsula, Pechora River Delta, Bolshezemelskaya Tundra, Polar Ural Mountains, Yamal Peninsula, Lena River Delta, Indigirka River Delta, and Magadan District.

## 3. Results

### 3.1. Fauna Composition and Species Richness

Thirty-three Cladocera species (33 of order Anomopoda, 1—Onychopoda and 2—Ctenopoda) and 45 Copepoda species (8 Calanoida, 17 Cyclopoida, 20 Harpacticoida) and taxa were identified in the studied water bodies of Putorana (species list see in Appendix A). Two of Cladocera crustaceans had not previously been recorded from the region: *Eurycercus pompholygodes* Frey, 1975 and *Biapertura sibirica* (Sinev, Karabanov et Kotov, 2020). New findings for the Putorana Plateau are numerous among Copepoda (21 species). The largest number of species noted for the first time (15) belong to the order Harpacticoida, of which two species from the genera *Moraria* and *Bryocamptus* (family Canthocamptidae) are new to science. Unexpectedly, in the freshwater lakes of the central part and the western slopes of the plateau, two species typical for brackish waters are found: *Onychocamptus mochammed* (Blanchard & Richard, 1891) (family Laophontidae) and *Pseudobradia arctica* (Olofsson, 1917) (family Ectinosomatidae). A specific feature of harpacticoid fauna of the Putorana Plateau is a very high diversity of the families Canthocamptidae, Ectinosomatidae, Laophontidae, Parastenocaridae, Phyllognathopodidae (see Appendix A), which is not typical for freshwater regional fauna. Within the Cyclopoida order, four species are new to the region: *Eucyclops* cf. *arcanus* Alekseev, 1990, *E. speratus* (Lilljeborg, 1901), *Cyclops sibiricus* Lindberg, 1949 and *Diacyclops bisetosus* (Rehberg, 1880). Two species of the order Calanoida from the genera *Acanthodiaptomus* and *Mixodiaptomus* (family Diaptomidae) were new for science. Thus, considering new findings, the performed studies expanded the data on the microcrustacean fauna of the Putorana Plateau up to 125 species, by 22% of the previously known list of fauna [18].

Crustacean diversity in the studied water bodies was high: 18.7 species on average (ranging from 8 to 26). At the same time, the average number of crustaceans in one water body in the plankton was $15.9 \pm 4.3$ species, and that in the meiobenthos was $7.1 \pm 3.6$. The most common species in the studied water bodies were the cladocerans *Chydorus* cf. *sphaericus* (O.F. Müller, 1785), *Daphnia* cf. *longispina* O.F. Müller, 1776 and the copepod *Heterocope appendiculata* Sars, 1863) and *Megacyclops viridis* (Jurine, 1820). They each occurred in more than 22 localities (73–87%). The species *Bosmina* cf. *longispina* Leydig, 1860, *Acroperus harpae* (Baird, 1834), *Biapertura sibirica* (Sinev, Karabanov et Kotov, 2020), *Polyphemus pediculus* (Linnaeus, 1758), *Macrocyclops albidus* (Jurine, 1820) and *Acanthocyclops capillatus* (Sars, 1863), were also quite frequent in the samples and occurred in 47–63% of the investigated water bodies. Most of the species (52) of total species list were rare and occurred only in 1–5 water bodies.

It is characteristic that the species richness of Cladocera was highest in the foothill areas (basin of Lake Keta) and gradually decreased towards the central part of the plateau

(Table 1). At the same time, the species richness of Copepoda was much higher in mountainous areas (for example, Calanoida and Harpacticoida 2–2.5 times) than in a slightly hilly foothill zone. The number of microcrustacean species did not decrease along with an increase in altitude, but on the contrary increased in the reservoirs of the western slopes of the plateau (average height 347 m a.s.l.), reaching 62 species. It is characteristic that the species richness of the fauna of a separate reservoir in the foothills was noticeably higher (22.8) than in mountainous areas (17.8–17.9). This fact indicates a high heterogeneity in the distribution of species between the water bodies of the mountain valleys of the Putorana Plateau.

**Table 1.** Main characteristics of crustacean fauna from three areas of the Putorana Plateau in August 2021.

| Features | District (Medial Altitude m a.s.l.) | | |
|---|---|---|---|
| | Central Part (492 m) | Western Slopes (347 m) | Foothills (84 m) |
| Species richness of Cladocera | | | |
| Anomopoda | 23 | 24 | 28 |
| Ctenopoda | 1 | 1 | 2 |
| Onychopoda | 1 | 1 | 1 |
| Species richness of Copepoda | | | |
| Calanoida | 4 | 5 | 2 |
| Cyclopoida | 9 | 16 | 10 |
| Harpacticoida | 11 | 15 | 6 |
| Complex characteristics of fauna | | | |
| Cladocera/Copepoda ratio | 1.04 | 0.72 | 1.70 |
| Species number per water body | 17.9 | 17.8 | 22.8 |
| Total species number | 50 | 62 | 49 |

*3.2. Patterns in Species Richness and Assemblage Structure of Meiobenthic and Planktonic Crustacean*

The DistLM analysis showed that general species richness and abundances of species from different taxonomical and ecological groups of crustaceans depend on such environmental factors, as an area size and type of water supply of the water body, macrophytes species composition, altitude above the sea level and research area (Table 2). For meiofauna, the most significant factors were the type of water supply of the reservoir (rain, spring, mixed) and the research district. In total, these factors explained about 19% of the variability in the structure of species assemblages. The most important factors for the planktonic fauna were the size of the reservoir and the composition of aquatic plants, which also explained 19% of the total variability.

For meiobenthos the ordination axes dbRDA1 and dbRDA2 explain a small proportion of the total variation—14.2% and 12.9%, respectively (Figure 3a). Major defining factor, the district of the research, is correlated to the first axis, while the type of water supply is correlated to the second axis. For zooplankton, the size of water body as the main determining factor is correlated with the first axis explaining 19.6% of variations (Figure 3b). A significant proportion of the variations in the assemblages' structure remains unexplained, which is determined by the high heterogeneity of biotopes, the specifics of individual reservoirs and, as a result, the presence of a number of factors not taken into account in the analysis.

**Table 2.** The results of marginal and sequential tests of DistLM (AIC criterion, step-wise selection). Significant factors are in bold ($p < 0.02$).

| Group | $R^2$ | P | Prop. | Cumul. |
|---|---|---|---|---|
| **Meiobenthic crustacean** | | | | |
| Marginal Test | | | | |
| **District** | | **0.001** | **0.11151** | |
| Type | | 0.733 | 0.026129 | |
| **Feed** | | **0.002** | **0.083758** | |
| Area | | 0.47 | 0.032965 | |
| **Altitude** | | **0.003** | **0.092905** | |
| Flow | | 0.175 | 0.047264 | |
| Temperature | | 0.479 | 0.03343 | |
| pH | | 0.367 | 0.038716 | |
| ppm | | 0.022 | 0.06709 | |
| Sediments | | 0.31 | 0.040733 | |
| Macrophytes | | 0.212 | 0.044731 | |
| Fish | | 0.349 | 0.038301 | |
| Sequential test | | | | |
| **+DIST** | **0.11151** | **0.001** | **0.11151** | **0.11151** |
| **+FEED** | **0.19577** | **0.004** | **0.084269** | **0.19577** |
| Best Solution | | | | |
| SUM | 0.50756 | | | 0.50756 |
| **Planktonic crustacean** | | | | |
| Marginal Test | | | | |
| District | | 0.286 | 0.040146 | |
| Type | | 0.42 | 0.035657 | |
| Feed | | 0.216 | 0.044911 | |
| **Area** | | **0.001** | **0.10377** | |
| Altitude | | 0.547 | 0.030024 | |
| Flow | | 0.522 | 0.031115 | |
| Temperature | | 0.055 | 0.060912 | |
| pH | | 0.426 | 0.034961 | |
| ppm | | 0.113 | 0.049079 | |
| Sediments | | 0.229 | 0.042521 | |
| Macrophytes | | 0.024 | 0.075753 | |
| Fish | | 0.64 | 0.028557 | |
| Sequential test | | | | |
| **+AREA** | **0.10377** | **0.001** | **0.10377** | **0.10377** |
| **+MACR** | **0.19079** | **0.002** | **0.087018** | **0.19079** |
| Best Solution | | | | |
| SUM | 0.56154 | | | 0.56154 |

$R^2$—square root criterion by which significant factors in model are selected; **P**—probability of random influence of a factor; **Prop.**—the proportion of variability which explains each factor (in the marginal tests—without coaction of factors); **Cumul.**—running cumulative total (percent of the variability explained by the model.

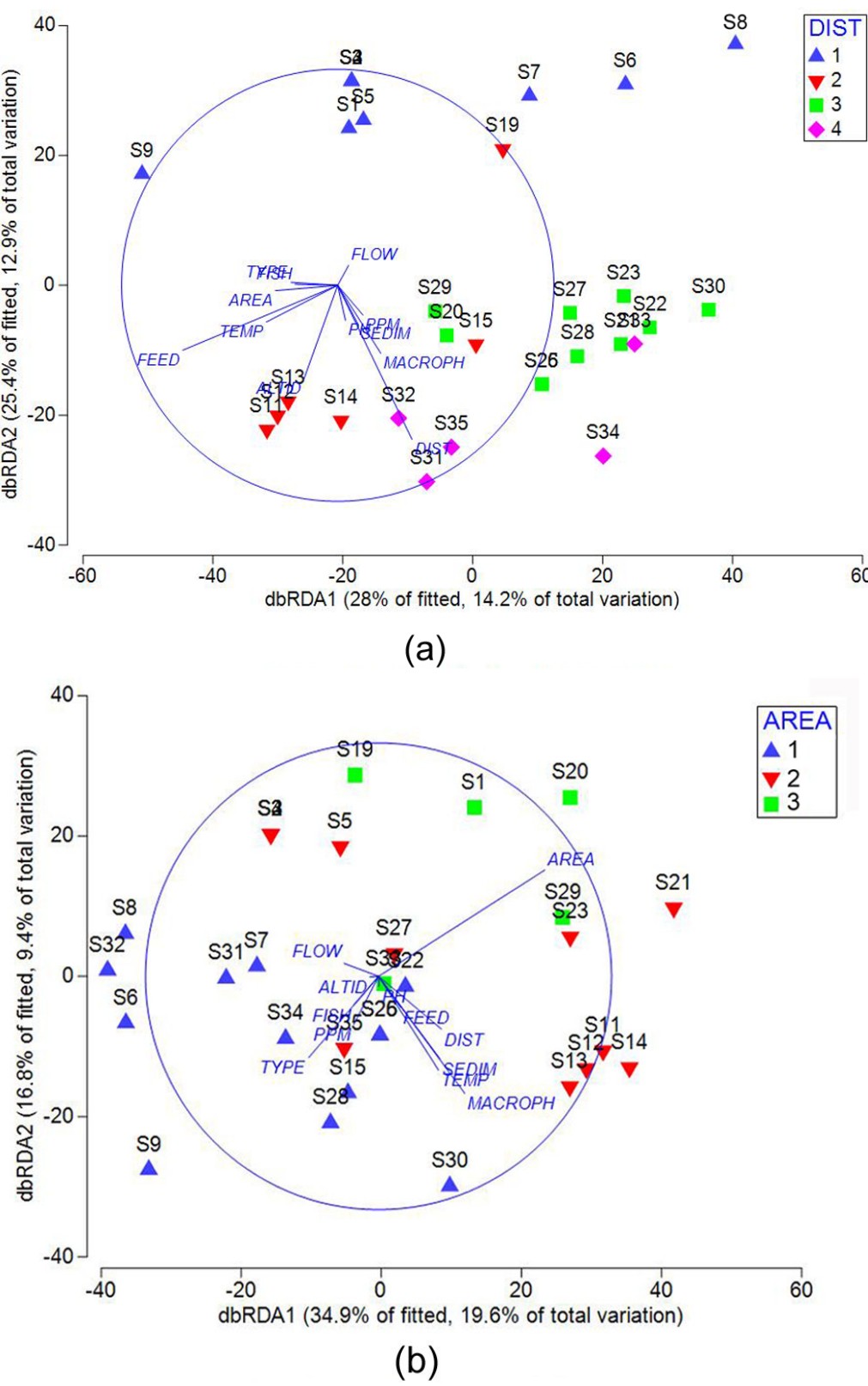

**Figure 3.** dbRDA ordination of microcrustacean assemblages from the Putorana Plateau. (**a**) Meio-fauna assemblages factored with different districts: 1—central part (Ayan Lake basin and Neral River valley); western slopes: 2—Burgul River valley, 3—Kutaramakan Lake basin and Irkinda River valley; 3—foothills (Keta Lake basin); (**b**) Zooplankton assemblages factored with different waterbodies sizes: 1—0.0001–0.009 km$^2$; 2—0.01–0.05 km$^2$, 3—0.15–7.7 km$^2$.

### 3.3. Characteristic and Differentiating Taxons

Among planktonic crustaceans, characteristic and differentiating species have been identified for water bodies with different water mirror areas (Tables 3 and 4). The most numerous of the characteristic species are dominants in species assemblages. All size classes of water bodies differ significantly in the structure of microcrustacean assemblages.

**Table 3.** Characteristic species of planktonic crustacean assemblages from waterbodies with different area size.

| Taxon | Average Abundance, % | Contribution to Explained Similarity, % | General Explained Similarity, % |
|---|---|---|---|
| **Assemblage 1. Area size 0.0001–0.009 km$^2$** | | | |
| *Heterocope appendiculata* | 6.22 | 6.61 | |
| *Bosmina* cf. *longispina* | 9.16 | 14.40 | |
| *Chydorus* cf. *sphaericus* | 11.67 | 20.53 | |
| *Daphnia* cf. *longispina* | 8.75 | 13.01 | 75.46 |
| *Pleuroxus truncatus* | 7.29 | 6.35 | |
| *Polyphemus pediculus* | 10.23 | 14.55 | |
| **Assemblage 2. Area size 0.01–0.05 km$^2$** | | | |
| *Heterocope appendiculata* | 20.80 | 29.49 | |
| *Acanthodiaptomus tibetanus* | 10.42 | 9.53 | 75.4 |
| *Bosmina* cf. *longispina* | 27.77 | 36.39 | |
| **Assemblage 3. Area size 0.15–7.7 km$^2$** | | | |
| *Heterocope appendiculata* | 42.05 | 81.54 | 81.54 |

**Table 4.** Differentiating species of planktonic crustacean assemblages from waterbodies with different area size (meaning of assemblage numbers in Table 3).

| Taxon | Contribution to Explained Difference, % | General Explained Difference, % |
|---|---|---|
| **Assemblages 1–2** | | |
| *Acanthodiaptomus tibetanus* | 6.34 | |
| *Heterocope appendiculata* | 12.32 | |
| *Alonella excisa* | 4.62 | |
| *Bosmina coregoni* | 4.42 | |
| *Bosmina* cf. *longispina* | 16.32 | |
| *Chydorus* cf. *sphaericus* | 6.68 | 73.81 |
| *Daphnia* cf. *longispina* | 5.87 | |
| *Pleuroxus truncatus* | 5.75 | |
| *Polyphemus pediculus* | 7.03 | |
| *Scapholeberis mucronata* | 4.46 | |
| **Assemblages 1–3** | | |
| *Heterocope appendiculata* | 22.57 | |
| *Cyclops scutifer* | 5.31 | |
| *Bosmina coregoni* | 4.20 | |
| *Bosmina* cf. *longispina* | 11.86 | |
| *Daphnia* cf. *longispina* | 5.20 | 68.7 |
| *Chydorus* cf. *sphaericus* | 6.26 | |
| *Pleuroxus truncatus* | 4.04 | |
| *Polyphemus pediculus* | 5.63 | |
| *Scapholeberis mucronata* | 3.65 | |

**Table 4.** *Cont.*

| Taxon | Contribution to Explained Difference, % | General Explained Difference, % |
|---|---|---|
| **Assemblages 2–3** | | |
| *Heterocope appendiculata* | 24.42 | |
| *Acanthodiaptomus tibetanus* | 6.71 | |
| *Cyclops scutifer* | 6.17 | |
| *Alonella excisa* | 3.14 | 69.2 |
| *Bosmina* cf. *longispina* | 21.10 | |
| *Daphnia* cf. *longispina* | 3.60 | |
| *Holopedium gibberum* | 4.08 | |

For meiobenthos, it was possible to identify characteristic species for different areas of the Putorana Plateau (Table 5). For the central part of the plateau, differentiating species are also identified, the abundance of which distinguishes its assemblage structure from the others (Table 6). The communities of the remaining areas differ weakly with no differentiating species statistically identified.

**Table 5.** Characteristic species of meiobenthic crustacean assemblages from waterbodies of different districts of the Putorana Plateau.

| Taxon | Average Abundance, % | Contribution to Explained Similarity, % | General Explained Similarity, % |
|---|---|---|---|
| **Assemblage 1. Central part** | | | |
| *Attheyella northumbrica trisetosa* | 26.57 | 51.86 | 76.16 |
| *Maraenobiotus insignipes* | 24.92 | 24.30 | |
| **Assemblage 2. Western slopes: Burgul River valley** | | | |
| *Attheyella northumbrica trisetosa* | 14.58 | 20.24 | |
| *Moraria mrazeki* | 9.06 | 9.48 | 50.26 |
| *Phyllognathopus paludosus* | 13.97 | 17.52 | |
| *Chydorus* cf. *sphaericus* | 10.70 | 12.50 | |
| **Assemblage 3. Western slopes: Kutaramakan Lake basin and Irkinda River valley** | | | |
| *Maraenobiotus insignipes* | 13.16 | 13.30 | |
| *Moraria mrazeki* | 10.12 | 25.84 | 68.04 |
| *Biapertura sibirica* | 8.21 | 13.79 | |
| *Chydorus* cf. *sphaericus* | 11.32 | 12.11 | |
| **Assemblage 4. Foothills** | | | |
| *Moraria mrazeki* | 15.94 | 17.06 | 38.52 |
| *Chydorus* cf. *sphaericus* | 16.48 | 21.46 | |

**Table 6.** Differentiating species of meiobenthic crustacean assemblages from central part of Putorana Plateau (meaning of assemblage numbers in Table 5).

| Taxon | Contribution to Explained Difference, % | General Explained Difference, % |
|---|---|---|
| **Assemblages 1–2** | | |
| *Attheyella northumbrica trisetosa* | 14.62 | |
| *Bryocamptus* sp. nov. | 5.49 | |
| *Epactophanes richardi* | 4.29 | |
| *Maraenobiotus insignipes* | 16.01 | |
| *Moraria mrazeki* | 5.22 | 67.01 |
| *Pesceus schmeili* | 7.14 | |
| *Phyllognathopus paludosus* | 8.05 | |
| *Chydorus* cf. *sphaericus* | 6.17 | |

**Table 6.** *Cont.*

| Taxon | Contribution to Explained Difference, % | General Explained Difference, % |
|---|---|---|
| **Assemblages 1–3** | | |
| *Attheyella northumbrica trisetosa* | 14.46 | |
| *Bryocamptus arcticus* | 3.14 | |
| *Bryocamptus* sp. nov. | 5.18 | |
| *Epactophanes richardi* | 5.31 | |
| *Maraenobiotus insignipes* | 16.07 | 68.01 |
| *Moraria mrazeki* | 5.51 | |
| *Pesceus schmeili* | 7.62 | |
| *Biapertura sibirica* | 4.47 | |
| *Chydorus* cf. *sphaericus* | 6.16 | |
| **Assemblages 1–4** | | |
| *Attheyella northumbrica trisetosa* | 13.87 | |
| *Bryocamptus* sp. nov. | 5.05 | |
| *Epactophanes richardi* | 3.94 | |
| *Maraenobiotus insignipes* | 13.22 | 59.83 |
| *Moraria mrazeki* | 8.43 | |
| *Pesceus schmeili* | 6.57 | |
| *Chydorus* cf. *sphaericus* | 8.75 | |

## 4. Discussion

### 4.1. Specificity of the Regional Fauna

The list of microcrustacean fauna of the Putorana Plateau, compiled based on the original and literature data, includes 62 species of Cladocera and 63 species of Copepoda. Such species richness is comparable to well-studied Arctic regions such as the Bolshezemel-skaya tundra (55 Cladocera and 62 Copepoda) [18], the Lena Delta (37 and 82) [35,44] and the Yamal peninsula (42 and 50) [38]. The remaining known fauna of microcrustaceans of the north Eurasia, both the Far East and Siberia, and the East European Plain, trailing on average 1.5–3 times the Putorana Plateau [33].

Thus, despite the harsh climate, the reservoirs of the plateau are inhabited by one of the richest microcrustacean fauna for the Arctic zone of Eurasia. For example, the number of Cladocera species (62) here is the largest among all known subarctic and arctic regional fauna. The number of noted here species of the Calanoida order (16) is comparable only to Western Siberia [18]. This fact confirms the previously noted tendency of increasing in number of species from the north of the East European Plain to the east to the regions of Siberia [17]. The species richness of the order Cyclopoida (27) of the Putorana Plateau does not stand out from other Arctic regions. However, the fauna of the order Harpacticoida is very rich in both families (5) and species (20) and is lesser only than in another region of Middle Siberia—the Lena River Delta (about 30 species) [18]. It is characteristic, that taxa new to science have been discovered on the plateau among both Calanoida and Harpacticoida. A number of new species of harpacticoids are also noted in the Lena River Delta [44]. What reasons could lead to an increase in the diversity of microcrustaceans in Middle Siberia?

The last glaciation was partial in the territory of Middle Siberia [5]. In particular, glacial domes on the Putorana Plateau lay only in the northwestern part of the highlands. Ice sheets did not reach the bottom of most large tectonic basins where the lakes were located and certainly did not fill them [5,45]. The lakes remained glacier-free, as evidenced by the continuous accumulation of lake bottom sediments [4]. Apparently, the high species richness of microcrustaceans is associated with the presence of Pleistocene fauna in the north of Middle Siberia. In addition, numerous findings of endemic Copepoda species on the Putorana Plateau and in the Lena River Delta may indicate the presence of the Harpacticoida and Calanoida speciation center in the region, which is rare in the far north.

The latter hypothesis requires verification, based on additional studies of the Copepoda fauna of the Taimyr Peninsula and the Anabar Plateau.

### 4.2. New Records for the Region

Most of the microcrustaceans discovered for the first time for the Putoran Plateau belong to meiofauna, which has been studied extremely poorly so far.

**Cladocera.** For the first time, two species of Anomopoda have been noted. *E. pompholygodes* is found in marshy water bodies of tundra and arctic deserts. Its area covers the subarctic and arctic regions of Eurasia, including the Taimyr Peninsula [29,45]. *B. sibirica* is a coldwater eurybiont species with a wide Palearctic range tending towards the polar zone [46].

**Copepoda**. Among meiobenthic Harpacticoida there are 15 new species for the region, of which two are new for science. Five species, *Attheyella nordenskioldii* (Lilljeborg, 1902), *Bryocamptus arcticus* (Lilljeborg, 1902), *B. krochini* (Borutzky, 1951), *Maraenobiotus brucei* (Richard, 1898), *Moraria duthiei* (T. Scott & A. Scott, 1896), tend to high latitudes and are often found in small tundra reservoirs [27]. One species *Bryocamptus vejdovskyi* (Mrázek, 1893) has a wide Holarctic range, found both in large lakes and in sphagnum bogs [24]. *Attheyella northumbrica trisetosa* (Chappuis, 1929) is Palearctic species typical for lakes of the forest and tundra [27]. Interstitial species *Elaphoidella gracilis* (Sars, 1863), *Parastenocaris brevipes* Kessler, 1913 and *Phyllognathopus paludosus* Mrázek, 1893 are typical for peat sphagnum bogs. The range of the first one covers European parts of Eurasia, and the latter two species are noted for both Eurasia and North America [24]. These three species have never been noted in the Subarctic and to the east of the Ural Mountains. Two species *Onychocamptus mochammed* (Blanchard & Richard, 1891) and *Pseudobradia arctica* (Olofsson, 1917), found in fresh waters of the plateau, are typical for brackish waters: estuaries and coastal lagoons [23]. Most likely, these species are invadive species brought here by birds of the orders Anseriformes and Charadriiformes from the estuary zone of the Yenisei River, where they were previously been noted [24]. Species new to science belong to the genera *Bryocamptus* and *Moraria* and differ from previously known species in the structure of the furcal branches and the armament of the thoracic limbs. Another species, *Moraria* cf. *mrazeki*, could not be uniquely correlated with any of the known species. However, due to the small number of individuals represented only by females, it is premature to distinguish it as new to science.

Four Cyclopoida species are new for the region. Two species of genus *Eucyclops* (*E. speratus* (Lilljeborg, 1901) and *E.* cf. *arcanus* Alekseev, 1990) belong to species complexes that include a large number of hardly identifiaed species [47]. *E. speratus* has a wide Palearctic area [23], while *E. arcanus* is known from Zabaykalye and northern Siberia [48]. *Cyclops sibiricus* Lindberg, 1949 is found in a number of regions of Eastern Siberia, the Far East, as well as northern Canada [23]. *Diacyclops bisetosus* (Rehberg, 1880) is a coldwater species with a range tending towards the northern regions [27].

Among Calanoida, our research has confirmed the plateau habitation of the relict species *Acanthodiaptomus tibetanus* (Daday, 1907) [16], which was previously called into question [28]. The finds of two new to science species of the Diaptomidae family deserve special attention. Species *Mixodiaptomus* sp. nov. with two pairs of lobes on the last segments of the thorax, close to *Mixodiaptomus incrassatus* (Sars, 1903) was found in the valley of the Neral River. In the water bodies of the vicinity of Lake Kutaramakan, *Acanthodiaptomus* sp. nov. was found, close to *A. tibetanus*, but having a smaller size and a different shape of the lobes of the last thoracic segment.

### 4.3. Crustacean Assemblage Structure and Regulating Factors

Assemblages of zooplankton, characteristic of the water bodies of different size clearly differ from each other in composition and abundance of dominant species (Table 3). Small and usually rare species contribute significantly less to the differences between assemblages than dominants (Table 4). Two factors played a key role in the formation of assemblages of

planktonic fauna—the water body surface size, indirectly associated with the hydrological type (lake, small lake, or puddle), and the composition of macrophytes. The influence of the size of the water body on the species richness of microcrustaceans has been already repeatedly noted [17,35,49], etc. The larger the size of the reservoir, the higher the diversity of microbiots that a specific fauna inhabits. However, often together with an increase in species richness in large lakes, the spectrum of dominants narrows, and the similarity of the composition of dominant species within the region increases, while small reservoirs show a large variety of assemblages structure [48]. Similar trends are noted for the reservoirs of the Putorana Plateau, when the number of dominant species for small lakes is six, and for large ones it is reduced to one main dominant (Table 3).

Assemblages of meiobenthos, identified for different areas of the Putorana Plateau, differ in the composition of characteristic species (Table 5), but it was possible to distinguish differentiating species only for the central mountain region of the plateau. Species *Maraenobiotus insignipes* (Lilljeborg, 1902), *Attheyella northumbrica trisetosa* (Chappuis, 1929) dominate in meiobenthos of the water bodies of the central part, while numerous in all other districts *Chydorus* cf. *sphaericus* (O.F. Müller, 1785) is completely absent. The bottom sediments of mountain lakes are depleted in plant detritus, which often adversely affects the abundance of Cladocera [49]. The most significant factors for the formation of meiobenthic assemblages were the research district, altitude above the sea level and the type of water supply of the reservoir. The influence of the research district factor on the distribution of fauna in the mountains is associated with the isolation of mountain valleys and the ability of organisms to disperse. The main taxonomic group of meiobenthos on the Putorana Plateau is Harpacticoida. These crustaceans living in the sediment layer settle very slowly, which demonstrates the depletion of their fauna on island territories [24,33]. Together with the research district, the characteristics of water bodies such as altitude and the type of water supply are also changing. In the central region of the plateau, lakes mainly have rain supply, and in the foothills—mixed or river supply [22]. Occasionally there are reservoirs with groundwater supply. Thus, the mineralized lakes of the Burgul River valley fed by groundwaters are clearly distinguished on Figure 3a. Interestingly, the fauna of meiobenthos is richer in the central part and on the western slopes of the plateau than in the foothills. Changes in the composition of the fauna in the direction deep into the plateau were noted earlier too [16]. This fact is probably explained not so much by the high-altitude gradient of environmental conditions, but by the high role of the eventuality of the species introduction into mountain reservoirs, compared to the foothills. Meiofauna of water body is formed by the species that were able to get into it, and the spectrum of these species is different for each reservoir, which contributes to a high variety of composition and structure of assemblages in mountainous areas. In addition, it is possible that high species richness is partly due to the specific structure of the ecosystems of the Putorana Plateau, preserved without significant changes in Pleistocene refugia.

*4.4. Biogeographical Position of the Putoran Plateau Fauna*

The known fauna of the Putorana Plateau has mixed character and composed of species from different biogeographical faunistic complexes (see Appendix A). The presented composition and characteristics of faunistic complexes are based on those described for Cladocera by Kotov A.A. [50] and further expanded for Copepoda [51]. In the total species list of microcrustaceans from the Putorana Plateau (based on original and literature date) cosmopolites and species with wide Palearctic ranges (18.5% and 42.7 of fauna, respectively) are the most diverse (Table 7). Representatives typical of the Subarctic and Arctic were also numerous and accounted for 17.7%, which emphasizes the belonging of the Putorana Plateau to the high latitudes. Holarctic species compose 8.1% of total fauna. Eastern Asian–North American species (*Daphnia* cf. *dentifera* Forbes, 1893, *Ophryoxus kolymensis* Smirnov, 1992, *Chydorus* cf. *biovatus* Frey, 1985 и *Cyclops sibiricus* Lindberg, 1949) in total made up 2.4% of the observed fauna. These species may be attributed as relics of the ancient Beringian refugium [50], a vast territory, that previously united Alaska, the Far East

of Eurasia, and is currently partially flooded. Species of the East Asian complex, common in Middle and Eastern Siberia, as well as in the Far East, accounted for 4.8%. For most of them (*Diaphanosoma pseudodubium* Korovchinsky, 2000, *Acanthodiaptomus tibetanus* (Daday, 1907) and *Neutrodiaptomus pachypoditus* (Rylov, 1925)) the Putorana Plateau is the northernmost point of their distributional ranges. Three more species, *E. gracilis*, *P. brevipes*, *P. paludosa* (2.4%), previously were known in Eurasia only from the Western Palearctic, the Putorana Plateau is also the extreme northern point of their areas. Four copepod species new for science (3.2% of the fauna) can be considered as endemic to the Putorana Plateau, although there is a possibility that further research into Siberia will expand their distribution. Based on the structure of the areas, the species of the last four groups (12.8% of the fauna) can be tentatively attributed to the Pleistocene relics preserved in the reservoirs of the Putorana Plateau during the last ice age.

**Table 7.** Biogeographical structure of microcrustacean fauna of Putorana Plateau.

| Faunistic Complex | Copepoda | | Cladocera | | Total Species List | |
|---|---|---|---|---|---|---|
| | Number | % | Number | % | Number | % |
| Cosmopolites | 13 | 21.0 | 10 | 16.1 | 23 | 18.5 |
| Wide Palearctic | 15 | 24.2 | 38 | 61.3 | 53 | 42.7 |
| Holarctic | 4 | 6.5 | 6 | 9.7 | 10 | 8.1 |
| Subarctic and Arctic | 19 | 30.6 | 3 | 4.8 | 22 | 17.7 |
| East Asian | 3 | 4.8 | 3 | 4.8 | 6 | 4.8 |
| East Asian—North American | 1 | 1.6 | 2 | 3.2 | 3 | 2.4 |
| West Palearctic | 3 | 4.8 | 0 | 0 | 3 | 2.4 |
| Endemic | 4 | 6.5 | 0 | 0 | 4 | 3.2 |

Copepod species of the Putorana Plateau have more diverse ranges than cladocerans. Majority of found Cladocera species are cosmopolites or widely distributed in the Palearctic, while all the endemics and the majority of subarctic species belongs to Copepoda (Table 7). Differences between these groups can be explained by their different dispersal ability [33], its seems that Purorana Plateau served as refugium for relict Copepoda species, but its Cladoceran fauna is not so specific.

Based on our original data and available literature sources, a comparative analysis of the composition of microcrustaceans in the northern regions of Eurasia showed a high similarity between the two regions of Middle Siberia (Figure 4). The Putorana Plateau fauna has 75 species sharing with the Lena Delta, which is 60% of the species richness. The Yamal Peninsula (Western Siberia), which is close in the species composition to the Putorana Plateau, includes only 38% of the total species. The remaining regions compared overlap with the plateau fauna by no more than 30% of the species list.

Thus, the fauna of microcrustaceans of the subarctic and Arctic zones of Middle Siberia has its own specifics and is significantly distinguished against the background of neighboring regions. These variability in faunal composition are connected to the current climatic differences and historical factor. Polar zone of Middle Siberia lies mostly northern than another compared Eurasian regions and has a much harsher continental climate. For example, the mean annual air temperature of the Taymyr peninsula varies from −12 °C to −15 °C, the Yamal peninsula from −8 °C to −12 °C [52]. On the one hand, the climatic factor limits the penetration of a number of boreal species into the Arctic zone of Middle Siberia, both from the east and from the west. On the other hand, the lack of cover glaciation in Middle Siberia gave a chance to preserve the rich relict fauna in mountain and seaside refugia.

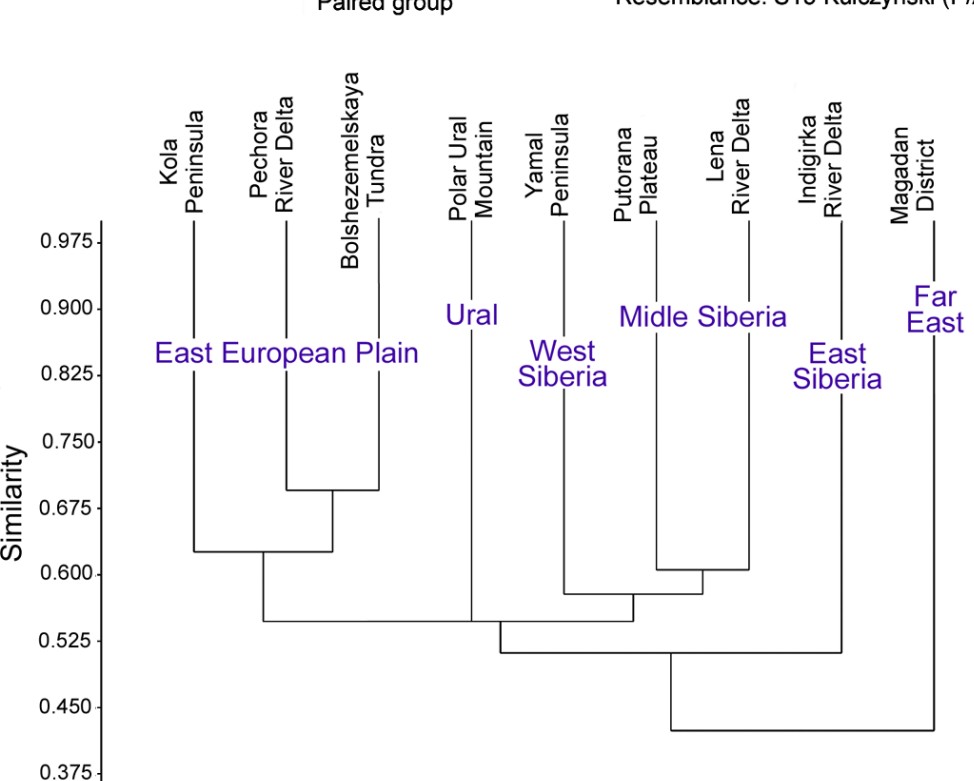

**Figure 4.** Dendrogram for hierarchical clustering of faunas of different regions of Subarctic and Arctic Eurasia regions.

## 5. Conclusions

1.  In the present study, 81 crustacean species were found in the water bodies of the Putorana Plateau: 45 Copepoda, 36 Cladocera. Two species of Cladocera and 21 species for Copepoda are new for the region. The list of known fauna of the region was increased by 22% of the previously known.
2.  The species richness of Copepoda (especially the meiobenthic Harpacticoida) is significantly higher in the central part and on the western slopes of the Putorana Plateau than in the foothills. The increase in the species richness of meiofauna in mountainous areas is associated with the high role of eventuality of species introducing into water bodies, which makes it possible to form different species composition and implement a different structure of assemblages in water bodies of the same region.
3.  Variations in the meiobenthic crustacean assemblage structure are due to the research district, type of water supply and, to a lesser degree, altitude above the sea level. The structure of planktonic crustacean assemblages was generally determined by the area size of the water body and less affected by the macrophytes composition.
4.  The total microcrustacean fauna of the Putorana Plateau consists of species belonging to different biogeographic faunistic complexes: 18.5% of fauna are cosmopolites; 42.7%—widely Palearctic; 17.7%—Subarctic and Arctic; 8.1%—Holarctic; 2.4%—Eastern Asian–North American; 4.8%—Eastern Asian; 2.4%—West Palearctic; 3.2%—endemics. About 12.8% of the species are relics that survived the last ice age in the reservoirs of the Putorana Plateau.
5.  The fauna of the northern part of Middle Siberia, in general, and the Putorana Plateau, in particular, is characterized by high species richness and significantly distinguishes from the fauna of both western and eastern regions of the Arctic. The specifics of faunal composition are connected to the climatic features of Middle Siberia and the preservation of Pleistocene fauna on its territory in refugia not subjected to the last glaciation.

**Author Contributions:** Conceptualization E.S.C. and A.A.N. (Anna A. Novichkova); methodology E.S.C., A.A.N. (Anna A. Novichkova), A.A.N. (Aleksandr A. Novikov), E.B.F., D.S.P. and L.V.V.; writing E.S.C. and A.A.N. (Anna A. Novichkova); revision E.S.C., A.A.N. (Anna A. Novichkova), A.A.N. (Aleksandr A. Novikov), E.B.F., A.I.G. and L.V.V. All authors have read and agreed to the published version of the manuscript.

**Funding:** The study of Cladocera was supported by the Russian Science Foundation (grant 18-14-00325); study of Copepoda was supported by the Russian Foundation for Basis Research (grant 20-04-00145); study of family Canthocamtidae was particularly supported by scientific research theme (N° 122040600025-2) of Institute of Biology of Komi Science Centre of the Ural Branch of the Russian Academy of Sciences.

**Institutional Review Board Statement:** Not applicable.

**Informed Consent Statement:** Not applicable.

**Data Availability Statement:** Not applicable.

**Acknowledgments:** The authors are grateful to the staff of the United Directorate of Taimyr Nature Reserves for help in organizing of the field works. Many thanks to M.G. Bondar' for his assistance with field sampling, searching literature about the Putorana Plateau and consultations in work with maps and photos. We are grateful to N.G. Sheveleva for consultation in Calanoida identification and to O.P. Dubovskaya for basic information about the microcrustacean fauna of the region. This research was performed according to the Development program of the Interdisciplinary Scientific and Educational School of M.V. Lomonosov Moscow State University "The future of the planet and global environmental change".

**Conflicts of Interest:** The authors declare no conflict of interest. The funders had no role in the design of the study; in the collection, analyses, or interpretation of data; in the writing of the manuscript, or in the decision to publish the results.

## Appendix A

**Table A1.** Species list, presence and faunistic complexes of crustaceans from plankton and meiobenthos in water bodies of three districts of the Putorana Plateau in August 2021. (*—species noted for the first time, **—species new to science; range types: ARC (P)—Subarctic and Arctic of Palearctica, ARC (C)—Circumarctic, C—cosmopolite or widespread unrevised species, EA—East Asian, EA-NA—East Asian—North American; END—endemic, HOL—Holarctic; wP—wide Palearctic, WP—West Palearctic).

| Taxon | District (Medial Altitude m a.s.l.) | | | Faunistic Complex |
|---|---|---|---|---|
| | Central Part (492 m) | Western Slopes (347 m) | Foothills (84 m) | |
| Class Branchiopoda Order Anostraca | | | | |
| Family Chirocephalidae *Polyartemia forcipata* Fischer, 1851 | + | | | ARC (P) |
| Subclass Cladocera Order Anomopoda | | | | |
| Family Bosminidae *Bosmina (Eubosmina) coregoni* Baird, 1857 | + | + | | wP |
| *B. (Eubosmina)* cf. *longispina* Leydig, 1860 | + | + | + | wP |
| Family Eurycercidae *Eurycercus (Eurycercus)* cf. *lamellatus* (O.F. Müller, 1776) | + | + | + | wP |
| * *E. (E.) pompholygodes* Frey, 1975 | | + | + | ARC (P) |

**Table A1.** *Cont.*

| Taxon | District (Medial Altitude m a.s.l.) | | | Faunistic Complex |
|---|---|---|---|---|
| | Central Part (492 m) | Western Slopes (347 m) | Foothills (84 m) | |
| Family Chydoridae | | | | |
| *Acroperus harpae* (Baird, 1834) | + | + | + | wP |
| *Alona guttata* Sars, 1862 | + | + | + | C |
| *A. intermedia* Sars, 1862 | | | + | wP |
| *A. quadrangularis* (O.F. Müller, 1785) | | + | | wP |
| *Alonella excisa* (Fischer, 1854) | + | + | + | wP |
| *A. exigua* (Lilljeborg, 1901) | | + | + | C |
| *A. nana* (Baird, 1850) | | | + | C |
| *Alonopsis elongata* (Sars, 1861) | + | + | | ARC (P) |
| *Biapertura affinis* (Leydig, 1860) | + | + | + | wP |
| * *B. sibirica* (Sinev, Karabanov et Kotov, 2020) | + | + | + | wP |
| *Chydorus* cf. *sphaericus* (O.F. Müller, 1785) | + | + | + | wP |
| *Coronatella rectangula* (Sars, 1862) | + | + | + | wP |
| *Flavalona costata* (Sars, 1862) | | | + | wP |
| *Graptoleberis testudinaria* (Fischer, 1851) | | | + | wP |
| *Pleuroxus trigonellus* (O.F. Müller, 1785) | + | + | + | wP |
| *P. truncatus* (O.F. Müller, 1785) | + | + | + | wP |
| *Pseudochydorus globosus* (Baird, 1843) | + | | + | HOL |
| Family Ophryoxidae | | | | |
| *Ophryoxus gracilis* Sars, 1862 | + | + | + | wP |
| *O. kolymensis* Smirnov, 1992 | + | + | + | EA-NA |
| Family Daphnidae | | | | |
| *Daphnia (Daphnia) cristata* Sars, 1862 | + | | + | wP |
| *D. (Daphnia)* cf. *dentifera* Forbes, 1893 | + | + | + | EA-NA |
| *D. (Daphnia) galeata* Sars, 1864 | | | + | C |
| *D. (Daphnia) longiremis* Sars, 1862 | + | | | HOL |
| *D. (Daphnia)* cf. *longispina* O.F. Müller, 1776 | + | + | + | wP |
| *D. (Daphnia) pulex* Leydig, 1860 | + | + | + | C |
| *Ceriodaphnia pulchella* Sars, 1862 | | + | | C |
| *Scapholeberis mucronata* (O.F. Müller, 1776) | + | + | + | wP |
| *Simocephalus expinosus* (De Geer, 1778) | | | + | C |
| *S. vetulus* (O.F. Müller, 1776) | + | + | + | wP |
| Order Ctenopoda | | | | |
| Family Sididae | | | | |
| *Sida ortiva* Korovchinsky, 1979 | | | + | EA |
| Family Holopediidae | | | | |
| *Holopedium gibberum* Zaddach, 1855 | + | + | + | C |
| Order Onychopoda | | | | |
| Family Polyphemidae | | | | |
| *Polyphemus pediculus* (Linnaeus, 1761) | + | + | + | wP |
| Subclass Copepoda Order Calanoida | | | | |
| Family Temoridae | | | | |
| *Heterocope appendiculata* Sars, 1863 | + | + | + | ARC (P) |
| *H. borealis* (Fischer, 1851) | | + | | ARC (P) |

**Table A1.** *Cont.*

| Taxon | District (Medial Altitude m a.s.l.) | | | Faunistic Complex |
|---|---|---|---|---|
| | Central Part (492 m) | Western Slopes (347 m) | Foothills (84 m) | |
| Family Diaptomidae | | | | |
| *Arctodiaptomus bacillifer* (Koelbel, 1885) | | + | | wP |
| *Acanthodiaptomus denticornis* (Wierzejski, 1887) | + | | + | HOL |
| *A. tibetanus* (Daday, 1907) | | + | | EA |
| ** *Acanthodiaptomus* sp. nov. | | + | | END |
| *Mixodiaptomus theeli* (Lilljeborg in Guerne & Richard, 1889) | + | | | wP |
| ** *Mixodiaptomus* sp. nov. | + | | | END |
| Order Cyclopoida | | | | |
| Family Cyclopidae | | | | |
| * *Eucyclops* cf. *arcanus* Alekseev, 1990 | + | + | + | EA |
| *E. denticulatus* (Graeter, 1903) | | + | | wP |
| *E. serrulatus* (Fischer, 1851) | + | + | + | C |
| * *E. speratus* (Lilljeborg, 1901) | | + | + | C |
| *Macrocyclops albidus* (Jurine, 1820) | + | + | + | C |
| *Paracyclops fmbriatus* (Fischer, 1853) | + | + | + | C |
| *Acanthocyclops capillatus* (Sars, 1863) | + | + | + | ARC (C) |
| *A.* cf. *robustus* (Sars, 1863) | | + | + | ARC (C) |
| *A. venustus* s. lat. (Norman & T. Scott, 1906) | | + | | ARC (P) |
| *A. vernalis* (Fischer, 1853) | | + | | C |
| *Cyclops scutifer* Sars, 1863 | | + | + | HOL |
| * *C. sibiricus* Lindberg, 1949 | | + | | EA-NA |
| *C. strenuus* Fischer, 1851 | + | + | | C |
| * *Diacyclops bisetosus* (Rehberg, 1880) | + | | | C |
| *D. nanus* (Sars, 1863) | + | + | | ARC (C) |
| *Megacyclops viridis* (Jurine, 1820) | + | + | + | C |
| *Microcyclops bicolor* (Sars G.O., 1863) | | + | + | C |
| Order Harpacticoida | | | | |
| Family Canthocamptidae | | | | |
| * *Attheyella northumbrica trisetosa* (Chappuis, 1929) | + | + | + | wP |
| * *At. nordenskioldii* (Lilljeborg, 1902) | | + | + | ARC (P) |
| * *Bryocamptus arcticus* (Lilljeborg, 1902) | | + | + | ARC (P) |
| * *B. krochini* (Borutzky, 1951) | | + | | ARC (P) |
| * *B. vejdovskyi* (Mrázek, 1893) | + | + | | HOL |
| ** *Bryocamptus* sp. nov. | + | | | END |
| *Canthocamptus glacialis* Lilljeborg, 1902 | + | | | ARC (P) |
| * *Elaphoidella gracilis* (Sars, 1863) | | | + | WP |
| *Epactophanes richardi* Mrázek, 1893 | + | + | | C |
| * *Maraenobiotus brucei* (Richard, 1898) | + | | | ARC (P) |
| *M. insignipes* (Lilljeborg, 1902) | + | + | | ARC (P) |
| * *Moraria duthiei* (T. Scott & A. Scott, 1896) | | + | | ARC (P) |
| *M. mrazeki* T. Scott, 1903 | + | + | + | ARC (P) |
| * *M.* cf. *mrazeki* | | + | | - |
| ** *Moraria* sp. nov. | | + | + | END |
| *Pesceus schmeili* (Mrázek, 1893) | + | + | | wP |
| Family Phyllognathopodidae | | | | |
| * *Phyllognathopus paludosus* Mrázek, 1893 | | + | | HOL |
| Family Parastenocaridae | | | | |
| * *Parastenocaris brevipes* Kessler, 1913 | + | + | | WP |

**Table A1.** *Cont.*

| Taxon | District (Medial Altitude m a.s.l.) | | | Faunistic Complex |
| --- | --- | --- | --- | --- |
| | Central Part (492 m) | Western Slopes (347 m) | Foothills (84 m) | |
| Family Ectinosomatidae<br>* *Pseudobradya arctica* (Olofsson, 1917) | | + | | ARC (C) |
| Family Laophontidae<br>* *Onychocamptus mochammed* (Blanchard & Richard, 1891) | + | | | C |

[1]—Arctic taxa also include species which ranges tending to high latitudes, although there are populations living more to south.

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
