# Peer review of "Assemblages of Meiobenthic and Planktonic Microcrustaceans (Cladocera and Copepoda) from Small Water Bodies of Mountain Subarctic (Putorana Plateau, Middle Siberia)"

_diversity, doi:10.3390/d14060492_

Round 1
Reviewer 1 Report
The paper deals with microcrustacean fauna of one of the most remote regions of Russia, Putorana Plateau. The information of freshwater fauna of this vast region is very scarce, and the paper is an important contribution to the study of region biodiversity, in my opinion, it will be widely cited and should be published. Importantly, the authors are experts in microcrustatian of Arctic-Subarctic region of Eurasia, and their species identification, the basis for such studies, is doubtless reliable.
The paper is of high quality, methods are well-described and sound. statistical methods are correctly applied, conclusions are well-supported. As far as I know, authors cite all existing literature on local faunas.
I have only two significant remarks
Lines 282-290
In table “Characteristic species of meiobenthic crustacean assemblages from waterbodies of differ-287 ent districts of the Putorana Plateau.” authors, together with species, list “Harpacticoida copepodites” and “Cyclopoida copepodites” as characteristic species for crustaceans assemblages. In my opinion, it is a wrong approach, as copepodite is not a species, but a stage, copepodites cannot be identified to species level, copepodites from on sample can belong to several species, etc. and so, should be excluded from analysis! Presence of copepodites is not a characteristic of species assemblage, but of the seasonal status of community…
Conclusion four 4 (lines 477-481).
“The microcrustacean fauna of the Putorana plateau consists of species belonging to different biogeographic faunistic complexes: 35% of fauna are cosmopolites; 34%—widely Palearctic; 18%—Subarctic and Arctic; 4%—Eastern Asian–North American; 4%— Eastern Asian–Far Eastern; 2%—West Palearctic; 3%—endemics. About 13% of the species are relics that survived the last ice age in the reservoirs of the Putorana plateau”
It will be very good to list species number as well as percentage.
Also, while authors discuss the biogeography of some species in discussion part, for many of species such characteristiс is not provided! It will be very convenient for the readers to add one more column into appendix table, with abbreviation of biogeographic characteristic of species ( C – cosmopolites, WP – widely Palearctic…, etc )
Also, I strongly suggest that percentage of faunistic composition should be counted separately for Cladocera and Copepoda. As I see from the table, majority of Cladocera are cosmopolites or widely deistributed Palaeoarctic species, while all endemics and majority of subarctic species belongs to Copepoda. Differenses between these group can be explained by different dispersal ability, its look like that Purorana Plateau served as refugium for relict endemic copepoda species, but its Cladoceran fauna is not so specific.
Minor correction
Lines 197-198
“Five of Cladocera crustaceans had not previously been recorded from the region: 197 Eurycercus pompholygodes Frey, 1975 and Biapertura sibirica (Sinev, Karabanov et Kotov, 2020).”
Obviously, there are only two species!
Line 350
...by birds of the orders anseriformes and charadriiformes…
Orders’ names should be capitalized.
Author Response
Thank you very much for positive reviewing of our manuscript!
Answers on reviewer's comments:
> I have only two significant remarks
> Lines 282-290
>In table “Characteristic species of meiobenthic crustacean assemblages from waterbodies of differ-287 ent districts of the Putorana Plateau.” authors, together with species, list “Harpacticoida copepodites” and “Cyclopoida copepodites” as characteristic species for crustaceans assemblages. In my opinion, it is a wrong approach, as copepodite is not a species, but a stage, copepodites cannot be identified to species level, copepodites from on sample can belong to several species, etc. and so, should be excluded from analysis! Presence of copepodites is not a characteristic of species assemblage, but of the seasonal status of community…
Answer: We are agree with reviewer. Information about copepodites was deleted.
>Conclusion four 4 (lines 477-481).
“The microcrustacean fauna of the Putorana plateau consists of species belonging to different biogeographic faunistic complexes: 35% of fauna are cosmopolites; 34%—widely Palearctic; 18%—Subarctic and Arctic; 4%—Eastern Asian–North American; 4%— Eastern Asian–Far Eastern; 2%—West Palearctic; 3%—endemics. About 13% of the species are relics that survived the last ice age in the reservoirs of the Putorana plateau”
It will be very good to list species number as well as percentage.
Answer: Table with this information was added in new Table 7.
> Also, while authors discuss the biogeography of some species in discussion part, for many of species such characteristiс is not provided! It will be very convenient for the readers to add one more column into appendix table, with abbreviation of biogeographic characteristic of species ( C – cosmopolites, WP – widely Palearctic…, etc )
Answer: It was added in the Appendix table.
> Also, I strongly suggest that percentage of faunistic composition should be counted separately for Cladocera and Copepoda. As I see from the table, majority of Cladocera are cosmopolites or widely deistributed Palaeoarctic species, while all endemics and majority of subarctic species belongs to Copepoda. Differenses between these group can be explained by different dispersal ability, its look like that Purorana Plateau served as refugium for relict endemic copepoda species, but its Cladoceran fauna is not so specific.
Answer: Table with this information was added to the text in new Table 7.
> Minor correction
>Lines 197-198
>“Five of Cladocera crustaceans had not previously been recorded from the region: Eurycercus pompholygodes Frey, 1975 and Biapertura sibirica (Sinev, Karabanov et Kotov, >2020).”
>Obviously, there are only two species!
Answer: Yes, it was corrected.
> Line 350
>...by birds of the orders anseriformes and charadriiformes…
>Orders’ names should be capitalized.
Answer: It was corrected.
Reviewer 2 Report
I can recommend to publish the MS sfter a minor revision. See the file with comments

Author Response
Thank you very much for positive reviewing of our manuscript!
Answers on reviewer comments from the pdf-file:
1. All comments about the orthography, style, format and taxonomical abbreviations of the article were corrected.
2. >In the MPDI, the fragment numbers are usually given ON photos, not out the photos.
Answer: It was corrected.
3. >"shape" ???
Answer: The term "shape" is widely used for reservoirs and water bodies in the sense of "form".
4. >Only one digit for abbreviation EVERYWHERE!
Answer: It was corrected through the text.
5. > Reference 38 is about Bykovsky Peninsula.
Answer: No, this reference about Shokalsky Island, which is close to Yamal and Gydanskiy Peninsulas.
38. Novichkova, A.A.; Chertoprud E.S. Cladocera and Copepoda of Shokalsky Island: new data from northwest Siberia. J. Nat. Hist. 2017, 51, 1781–1793. https://doi.org/10.1080/00222933.2017.1355077
6. > Add reference Bekker et al., 2012
Answer: It was added.
7. > to Maraenobiotus insignipes (Lilljeborg, 1902): Full species names are necessary only at first record in the text
Answer: It was first record in the text of this species. Before was only Maraenobiotus brucei (Richard, 1898).
8. > Add reference Bekker et al., 2012
Answer: It was added.